# A Database of Lung Cancer-Related Genes for the Identification of Subtype-Specific Prognostic Biomarkers

**DOI:** 10.3390/biology12030357

**Published:** 2023-02-24

**Authors:** Yining Liu, Min Zhao, Hong Qu

**Affiliations:** 1The School of Public Health, Institute for Chemical Carcinogenesis, Guangzhou Medical University, Guangzhou 510180, China; 2School of Science, Technology and Engineering, University of the Sunshine Coast, Maroochydore, QLD 4558, Australia; 3Center for Bioinformatics, State Key Laboratory of Protein and Plant Gene Research, School of Life Sciences, Peking University, Beijing 100871, China

**Keywords:** lung cancer, database, genetic, subtype, systems biology, biomarker

## Abstract

**Simple Summary:**

We developed a lung cancer-specific database containing genetic and literature data from over 10,000 separate studies. The cancer subtype information was meticulously curated and quality controlled, while the subtype-specific genetics can be explored in a novel manner. In addition, we created the Lung Cancer Gene (LCGene) database, an open-access web interface that enables researchers and clinicians to explore these data and conduct large-scale integrative analyses. On LCGene, users can perform gene list-based data integration to gain a quick understanding of the shared and unique characteristics of various subtypes of lung cancer. In summary, data from subtype-based survival analysis, comparative analysis, and CRISPR knockout provide additional novel information for genome-wide gene/biomarker screening in lung cancer subtypes.

**Abstract:**

The molecular subtype is critical for accurate treatment and follow-up in patients with lung cancer; however, information regarding subtype-associated genes is dispersed among thousands of published studies. Systematic curation and cross-validation of the scientific literature would provide a solid foundation for comparative genetic studies of the major molecular subtypes of lung cancer. Here, we constructed a literature-based lung cancer gene database (LCGene). In the current release, we collected and curated 2507 unique human genes, including 2267 protein-coding and 240 non-coding genes from comprehensive manual examination of 10,960 PubMed article abstracts. Extensive annotations were added to aid identification of differentially expressed genes, potential gene editing sites, and non-coding gene regulation. For instance, we prepared 607 curated genes with CRISPR knockout information in 43 lung cancer cell lines. Further comparison of these implicated genes among different subtypes identified several subtype-specific genes with high mutational frequencies. Common tumor suppressors and oncogenes shared by lung adenocarcinoma and lung squamous cell carcinoma, for example, exhibited different mutational frequencies and prognostic features, suggesting the presence of subtype-specific biomarkers. Our retrospective analysis revealed 43 small cell lung cancer-specific genes. Moreover, 52 tumor suppressors and oncogenes shared by lung adenocarcinoma and squamous cell carcinoma confirmed the different molecular mechanisms of these two cancer subtypes. The subtype-based genetic differences, when combined, may provide insight into subtype-specific biomarkers for genetic testing.

## 1. Introduction

Lung cancer (LC) is the leading cause of cancer-related deaths in both males and females, accounting for approximately 1.6 million deaths annually worldwide [1]. In the United States alone, there were roughly 130,180 LC-associated deaths in 2022. Small cell (SCLC) and non-small cell lung cancer (NSCLC) are the two main types [2]. NSCLC is the most common type, contributing to 80–85% of all LC cases and typically growing and metastasizing slower than SCLC. Despite ongoing efforts to develop effective treatments, the 5-year survival rate of NSCLC patients is only 14% for Stage IIIA, 5% for Stage IIIB, and 1% for Stage IV [3]. The most common types of NSCLC are adenocarcinoma, which constitutes over 30% of all diagnosed LC cases, and squamous cell carcinoma, which accounts for less than 30% [2]. Given the importance of these two LC subtypes, a genome-wide exploration of the molecular profile of lung adenocarcinoma (LUAD) and squamous cell carcinoma (LUSC) has been conducted by The Cancer Genome Atlas (TCGA) consortium [4]. Identification of the subtype is critical for developing effective treatment strategies for LC. There exist many clinical and histological methods to identify LC subtypes; however, molecular subtype information will independently provide reliable confirmation to guide precision medicine.

As a heterogeneous disease, LC has complex molecular mechanisms for unrestrained cell growth, which may be caused by promoter methylation, dysregulated gene expression, and/or alterations in tumor suppressor genes and oncogenes [2]. Despite the focus of thousands of publications on LC in patients with various subtypes of the disease, no literature collection-based effort has been performed to scrutinize the common and unique genetic information for each LC subtype. Furthermore, the majority of functional or clinical studies are single gene-based and fail to provide a complete picture of tumorigenesis for various cancer subtypes. Here, we created the LCGene database to provide a global genetic representation of various LC subtypes and a reusable genetic resource for LC supported by links to scientific literature-based evidence.

The gene-centered database, consisting of curated genes, will be useful for prioritizing genes based on their LC-associated relevance and identifying common and unique cellular events in different LC subtypes. Our data may be a solid starting point for the performance of a meta-analysis of the molecular mechanisms underlying different types of LC, and our curated literature collection may be an important indicator of the well-studied genes in specific types. Taken together, these data will serve as a foundational computational resource for LC biomarker discovery and validation.

## 2. Materials and Methods

### 2.1. Literature Search and Curation

The principal aim of this database is to aid LC research by maintaining a high-quality gene list that serves as a comprehensive, fully classified, richly, and accurately annotated platform with extensive cross-references and querying interfaces that are freely accessible to the scientific community. As shown in the workflow (Figure 1), the LC-implicated gene collection is primarily based on the Gene Reference Into Function (GeneRIF) database [5]. As previously described, we performed a keyword-based query against the GeneRIF database using Perl regular expression to extract relevant sentences: [(lung OR pulmonary) AND (cancer OR tumor OR carcinoma)] on 22 January 2021 [6]. In total, we found thousands of short sentences that were related to LC in 15,964 unique PubMed abstracts. The abstracts were then downloaded in Medline format and parsed into free text for manual curation.

Similarly, model species names such as mouse and rat were used to match species information. We mapped all non-human gene homologs to their human counterparts. Finally, we standardized all the gene descriptions by mapping them to the NCBI Entrez gene database [7], which will be useful for gene ID-based annotations. For example, the COX-2 gene is a common synonym for the PTGS2 gene; therefore, we saved PTGS2 and its associated gene ID (5743, available online: https://www.ncbi.nlm.nih.gov/gene/5743) (accessed on 12 February 2023) in our database.

Further literature curation included: (i) clustering of abstracts; (ii) extracting matched cancer subtypes; (iii) gathering species information; and (iv) formalizing gene symbols. Specifically, we first performed semantic similarity clustering based on sentence-keyword matching, which is useful for curating information regarding the same gene from multiple sources of evidence. We only included the records if the manuscript clearly stated that they were “relevant” to the disease and their manifestations included mutations, up- or downregulation of expression, changes in copy number, or gene fusion.

To control data quality, literature with unclear and confusing conclusions were deemed insufficient for inclusion in the database. In total, we collected 10,960 PubMed abstracts that were associated with LC genes. The curated sentences were then searched using LC subtype keywords such as LUAD, LUSC, and NSCLC to extract the appropriate subtype information. For example, in the sentence “stabilization of surviving may contribute to the apoptosis resistance effect of COX-2 in non-small cell lung cancer”, the cancer subtype could be assigned as “non-small cell lung cancer”.

### 2.2. Biological Annotation and Pre-Computed Data 

To provide insight into the biology of the collected genes, we retrieved comprehensive biological functional annotations from public resources. Firstly, basic gene information was retrieved using curated gene IDs, including the gene alias, and crosslinked to genome databases such as Ensembl [8]. The functional annotations were further extracted from gene ontology annotation (GOA) [9] and the Kyoto Encyclopedia of Genes and Genomes (KEGG) database [10] for pathway mapping. The tumor suppressors and oncogenes were obtained from the TSGene 2.0 [11] and ONGene [6] databases. These various gene sets will aid gene categorization into tumor suppressors and oncogenes.

In addition to annotations, we retrieved all gene expression data from the BioGPS database [12]. Co-expressed protein-coding and long non-coding RNA genes for the TCGA LC dataset was based on the LnCaNet database [13], which provides a foundation for the genome-wide investigation into potential long non-coding RNAs that are associated with subtype-specific genes based on evidence from the scientific literature. Differential expression based on large-scale TCGA data was also calculated for bulk download. Differentially expressed genes from TCGA LUAD and TCGA LUSC will be useful for exploring changes at the mRNA transcript level, which may be overlooked by single-gene studies.

By mapping to the Catalogue of Somatic Mutations in Cancer (COSMIC) database [14], we presented all the somatic mutations for 2507 LC-related genes. To determine potential interacting partners, we integrated regulatory interactors from the Transcription Factor (TRANSFAC) [15] and Pathway Commons [16] databases. We subsequently mapped these 2507 genes to the most recently updated CRISPR knockout data in all 43 LC cell lines [17], thus providing potential gene editing sites.

To provide an overview of the potential gene regulation, we also incorporated the experimental validated gene regulation data, including the experimentally validated and predicted competing endogenous RNA (ceRNA) from LncACTdb 3.0 [18].

Gene ontology (GO) functional enrichment analysis was conducted through the g:Profiler online server [19]. Corrected *p*-values, calculated using all human protein-coding genes as the baseline, were used to rank the enriched functional annotations of input genes. The cBioPortal, an online server for visualization and analysis of multidimensional cancer genomics data, was used for all sample-based mutational analyses [20]. Next, we integrated additional functional information, including the metabolic network and transporters [21].

### 2.3. Database Construction and Web Interface Coding

All the curated literature and annotation information were saved and stored in a relational database managed by the MYSQL system (available online: https://www.mysql.com/) (accessed on 12 February 2023). Specifically, for each annotation, we created a separate table to store the data. Server-side programming was implemented by Perl CGI to connect with MYSQL and print the HTML pages. To present our data, we created responsive websites using the Bootstrap framework (available online: https://getbootstrap.com/) (accessed on 12 February 2023). We used JavaScript library HighCharts (available online: https://www.highcharts.com/) (accessed on 12 February 2023) to summarize and visualize the data interactively by connecting them to the MYSQL database. For example, a word cloud containing a set of keywords was created to display the retrieved information, with the size and placement determined by how frequently they were mentioned in the literature. Similarly, HighCharts was used to visualize protein-protein interaction data as a network and to summarize various data categories on browsing pages.

## 3. Results and Discussion

We consolidated 2507 lung cancer (LC)-associated genes, including 2267 protein-coding and 240 non-coding genes (Appendix A). Among the 2507 LC-implicated genes, 1706 genes (68%) were curated with subtype information. We identified 71 genes based on the curated literature, with over 30 supporting abstracts from PubMed. Only one literature reference supported 769 (or 30.67%) of the 2507 LC-implicated genes, requiring further functional experimental validation. We identified 232 SCLC-related genes and 1324 NSCLC-related genes, which are the major subtypes of LC. For LUAD and LUSC, 535 and 251 genes, respectively, were curated.

### 3.1. Web Interface

Based on a systematic survey of LC-implicated genes in the literature, we developed a web interface to make these annotations publicly available. All the LC-implicated genes can be explored using a web browser. The genomic distribution of all the genes were plotted on 24 chromosomes, with individual charts for browsing (Figure 2A). Users may also browse each LC subtype to access all the LC-implicated genes within that subtype (Figure 2A). When browsing, the number of literature citations are provided for each gene, informing the user of the importance of particular genes in LC development. Tumor suppressors and oncogenes are also listed for quick access, and colored KEGG pathway maps allow the exploration of genes based on biological functions (Figure 2A).

A quick search button at the top right-hand side of each webpage can be used to conduct queries using official gene symbols or Entrez Gene IDs (Figure 2A). Advanced search functions are provided by typing in the gene name or its functional characteristics, including chromosome location, interacting partners, biological processes, or name of the associated disease. Users can search all the curated literature using keywords; this is useful when searching for a specific topic in biology. The results page will display a list of genes or literature that match the input keywords. The hyperlinked gene names or literature direct users to pages that contain information that is relevant to that gene. To organize the information for each gene, we divided our annotation details into five categories: general information, literature, gene expression, genetic mutation, and protein-protein interactions. When exploring the annotations, users can rapidly access any specific annotation by clicking on the labels at the top right-hand side of each web page (Figure 2B).

This database also facilitates the bulk download of all the curated genes, which is useful for any advanced integrative studies. Moreover, we identified 8224 differentially expressed events for the 641 LC-implicated genes in 107 matched cancer and normal sample pairs in the two TCGA datasets (Appendix A, LUAD; Appendix A, LUSC), which will be useful for the reconstruction of co-expression-based regulatory networks. Furthermore, we present pre-computational results of 24,415 co-expressed pairs with protein-coding genes (Appendix A), and 361,417 co-expressed pairs with long non-coding RNAs based on the matched TCGA cancer samples (Appendix A). All these pre-computed data will be useful for large-scale functional screening.

The most exciting development in cancer biology is the potential of gene editing technology to revolutionize cancer therapy. For example, the results of a Phase 1 clinical trial to evaluate the safety and feasibility of CRISPR-Cas9 gene editing in three patients with advanced multiple myeloma and myxoid/round cell liposarcoma were recently published [22]. In that study, T lymphocytes were removed from the patients and CRISPR-Cas9 was used to disrupt three genes encoding the endogenous T cell receptor α chain (TRAC), TCR β chain (TRBC), and programmed cell death protein 1 (PDCD1), which substantially improved antitumor immunity. The first CRISPR-engineered cancer immunotherapies have also been reported for LC [23]. Interestingly, we found 607 LC-related genes that were mapped to CRISPR knockout information in 43 LC cell lines (Appendix A), which may provide a critical foundation for functional cancer screening.

### 3.2. Genes Shared by Different LC Subtypes

To investigate genetic heterogeneity in LC, we overlapped LC-implicated genes with different subtypes (Figure 3A) and found 43 SCLC-specific genes with two or more citations in the scientific literature. It is important to note that different literature may assign different subtypes to the same gene. Consequently, these highly overlapping genes may play consistent roles in numerous subtypes of LC. Among these genes, the mutational frequency in the EP300 gene was substantially different across three public SCLC and seven NSCLC datasets (Figure 3B). Phosphorylation-mediated EP300 downregulation enhances oncogene expression during interphase and decreases histone H3 acetylation during mitosis, accelerating SCLC development [19]. This demonstrates that subtype-specific gene mining with the LCGene database may identify highly and distinctly mutated genes in LC subtypes, facilitating exploration of the underlying molecular mechanisms.

We found 52 shared tumor suppressor genes and oncogenes between LUAD and LUSC by focusing on two subtypes of NSCLC (Figure 3C). The enriched functional annotations of input genes were ranked with corrected *p*-values that were calculated using all human protein-coding genes as the baseline. These 52 genes are critical for many cellular processes and are mainly localized to the nucleoplasm (corrected *p*-value = 3.728 × 10^−7^), intracellular organelle lumens (corrected *p*-value = 1.423 × 10^−5^), and plasma membrane-bound cell projections (corrected *p*-value = 2.410 × 10^−2^) (Figure 3D). Therefore, these genes are mainly related to protein kinase activity (corrected *p*-value = 4.669 × 10^−8^) and regulated by phosphorylation (corrected *p*-value = 3.589 × 10^−20^) involving intracellular signal transduction (corrected *p*-value = 1.446 × 10^−17^) and the regulation of intracellular transduction (corrected *p*-value = 8.225 × 10^−17^). These data show that the 52 genes are involved in a wide range of fundamental biological processes related to cell proliferation and cancer progression.

KEGG pathway analysis revealed additional important cancer- and pluripotency-related signaling pathways (Figure 3E) such as phosphatidylinositol 3-kinase (PI3K)-protein kinase B (Akt) (corrected *p*-value = 6.505 × 10^−12^), hypoxia inducible factor 1 (HIF-1) (corrected *p*-value = 4.713 × 10^−11^), signaling pathways regulating pluripotency (corrected *p*-value = 4.753 × 10^−8^), thyroid hormone (corrected *p*-value = 1.356 × 10^−7^), and advanced glycation end products receptor (RAGE) (corrected *p*-value = 1.201 × 10^−5^). Interestingly, epidermal growth factor receptor (EGFR) tyrosine kinase inhibitor resistance (corrected *p*-value = 2.522 × 10^−12^) and endocrine resistance (corrected *p*-value = 2.459 × 10^−14^) highlighted the likelihood of drug resistance for both NSCLC subtypes (LUAD and LUSC). A few genes that were associated with choline metabolism raised the possibility of a metabolism-based molecular diagnosis for LC (corrected *p*-value = 2.394 × 10^−12^). Finally, these 52 genes are involved in programmed death-ligand 1 (PD-L1) expression and the PD-1 receptor checkpoint pathway (corrected *p*-value = 2.302 × 10^−7^), which are critical in cancer immunotherapy.

We further conducted mutational analysis using the cBioPortal [18]. By examining the mutational frequency separately in LUAD and LUSC, we found that these 52 cancer driver genes have a distinct mutational frequency in the two subtypes of NSCLC (Figure 4A). There are four genes (PIK3CA, SOX2, PRKCI, TP63) with high amplification in LUSC as compared with LUAD (Figure 4B), all of which are localized to chromosome 3q [20]; however, the LUAD-specific Kirsten rat sarcoma virus (KRAS) driver gene is more likely to be co-mutated with serine/threonine kinase 11 (STK11) (Figure 4C), which has been used to define a novel subtype with distinct biology, immune profiles, and therapeutic vulnerabilities [21]. This suggests that some shared genes in different subtypes may be associated with a common mechanism; however, because of their vastly different mutational patterns, these genes may play entirely different roles in cancer progression. When combined, the subtype-specific information in our LCGene database may demonstrate both common and unique molecular properties.

### 3.3. Potential Prognostic Biomarkers between Different Subtypes of LC

To explore potential prognostic applications of curated LC genes, we merged the 52 driver genes that were shared by LUAD and LUSC with 44 LC datasets with survival outcomes from the prognostic PRECOG database. These 44 datasets were classified into LUAD, LUSC, or large cell carcinoma (LCC) subtypes, and PRECOG computed Z-scores for each gene to characterize the gene expression and clinical outcomes across multiple datasets. In general, a positive Z-score for a gene associated with a specific dataset indicates increased expression (adverse survival), whereas a negative Z-score indicates decreased expression (favorable survival). The heatmap of PRECOG Z-scores for the 44 LC dataset shows 51 of the 52 driver genes (Figure 5). The first cluster of 18 genes (e.g., TP53, TP63, EGFR, KRAS, and STAT3) had a prognostic Z-score less than −1.96 (equivalent to a two-tailed *p* < 0.05) in the 11 LUAD datasets, indicating significantly favorable survival. For 9 LUSC datasets, the other two clusters containing MYC, PIK3CA, BRCA2, CDKN1A, and MTOR were favorable towards survival. These distinct links to different subtypes of LC may provide supporting evidence of specific mechanisms related to patient survival.

In addition, we examined the survival characteristics of 9781 samples from 7590 lung cancer patients in 28 independent studies in order to provide an overview of these genes in the cancer cohort. Appendix A depicts 52 genes, including TP53 and KRAS, that are significantly associated with survival outcome (log rank test *p*-value: 1.130 × 10^−4^, Appendix A). Therefore, our PRECOG and survival data provide independent evidence that these genes play crucial roles in lung cancer patient survival.

MicroRNA (miRNA) dysregulation is associated with the occurrence and progression of cancer. We curated 204 LC-related miRNAs in our database, which when inhibited or overexpressed, regulate the expression of their target genes, inhibiting cancer cell proliferation or metastasis. miRNAs have tremendous potential as therapeutic targets in LC since they suppress expression that is derived from thousands of mRNA transcripts. We did not extend our analysis to those key regulators because of the limits of this study. Subtype-specific miRNA regulatory patterns may provide a novel approach to reversing key oncogenic phenotypes, such as epithelial–mesenchymal transition.

Long non-coding RNAs (lncRNAs) are recent and important non-coding transcripts in cancer biology. In our curated gene list, we found several pseudogene-based antisense lncRNAs. We pre-computed expression profiles for thousands of lncRNAs in LUAD and LUSC; however, only a small portion has been functionally characterized based on evidence from the scientific literature. Our pre-computed differential expression data between tumor and normal tissues will be invaluable in furthering our understanding of RNA-based regulatory mechanisms, such as competing endogenous RNAs (ceRNAs). Some subtype-specific regulatory mechanisms, such as the ceRNA model, may provide novel insight into the imbalance in stoichiometry between target mRNAs and their postulated binding sites on ceRNAs.

## 4. Conclusions

LC is the most common cancer in both males and females, accounting for over one-quarter (25%) of cancer-related deaths. The accurate identification of LC subtypes is critical for precise medical treatment since therapy largely depends on understanding biological pathways and regulatory mechanisms that are associated with different subtypes. The goal of our data curation was to assess the subtype-specific genes that are associated with prognosis. After curating 15,964 PubMed abstracts related to LC, we collected 2507 genes that were associated with various LC subtypes. There are 232 SCLC-related genes and 1324 NSCLC-related genes in our data collection. We curated 535 and 251 genes for LUAD and LUSC, respectively; however, only 52 oncogenes or tumor suppressor genes were shared by LUAD and LUSC, confirming that these two major LC subtypes share common molecular mechanisms. A few important signaling pathways shared by LUAD and LUSC were identified through additional functional and mutational analyses. Our prognostic analysis provides a global perspective on the heterogeneous genetic structures of different LC subtypes, which may aid patient survival. Our future direction is to focus further on the subtype-unique gene set, which may help us to understand the underlying disease mechanisms and identify novel therapies for specific LC subtypes.

## Figures and Tables

**Figure 1 biology-12-00357-f001:**
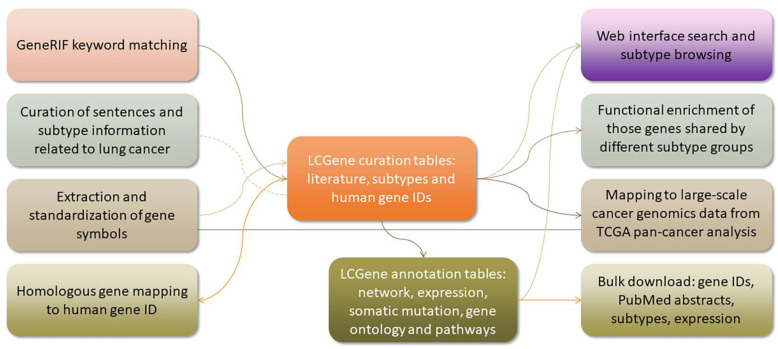
Literature curation and functionally annotated workflow for the LCGene database.

**Figure 2 biology-12-00357-f002:**
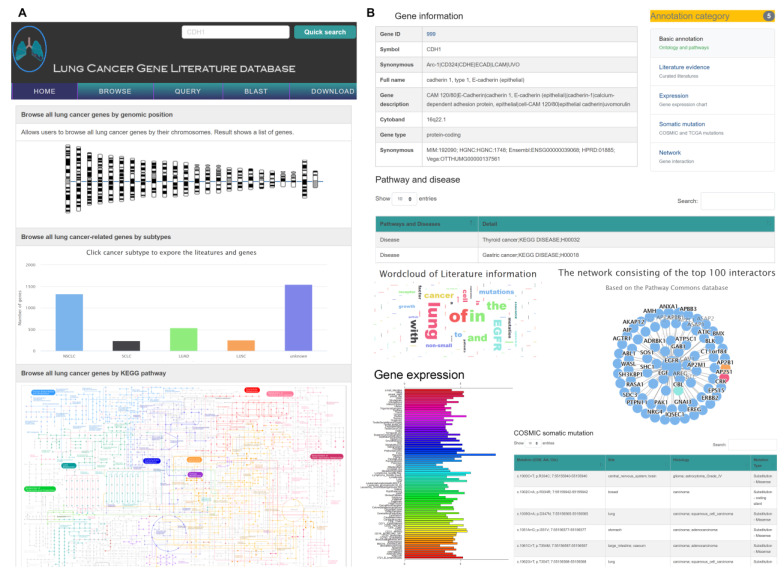
The LCGene database web interface. (**A**) Browsing the genes in the database using chromosomes, cancer subtype, literature support, tumor suppressors, oncogenes, and KEGG pathways. (**B**) Basic annotations for genes include the gene description, alias, crosslinks, gene ontology, pathways, and disease information. The associated literature is presented by a word cloud to highlight the key message for the gene of interest. Protein-protein interaction data from the Pathway Commons database are used to depict interacting partners. Gene expression patterns from the BioGPS database and somatic mutation data from the COSMIC database are presented for filtering and searching. KEGG, Kyoto Encyclopedia of Genes and Genomes.

**Figure 3 biology-12-00357-f003:**
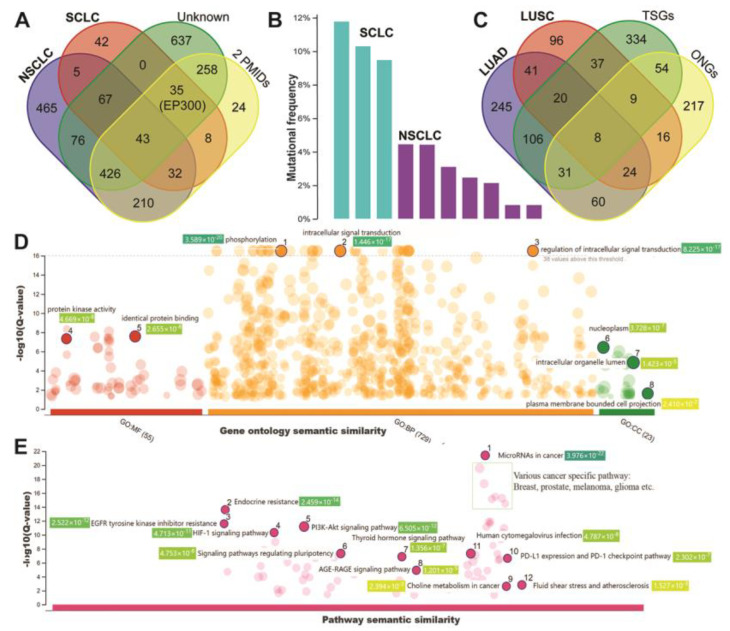
Overlapping and functional enrichment of genes that are associated with different subtypes. (**A**) Venn diagram of NSCLC, SCLC, and LC genes of unknown subtypes that were mentioned twice or more in the literature. (**B**) The mutational rate of EP300 in three SCLC and seven NSCLC studies. (**C**) Overlapping tumor suppressor genes (TSGs) and oncogenes (ONGs) with genes related to LUAD and LUSC. (**D**) Gene ontology enrichment analysis of the 52 oncogenic and tumor suppressor genes shared by LUAD and LUSC. (**E**) Enriched KEGG pathways for the 52 oncogenic and tumor suppressor driver genes shared by LUAD and LUSC. KEGG, Kyoto Encyclopedia of Genes and Genomes; LUAD, lung adenocarcinoma; LUSC, lung squamous cell carcinoma; NSCLC, non-small cell lung cancer; SCLC, small cell lung cancer.

**Figure 4 biology-12-00357-f004:**
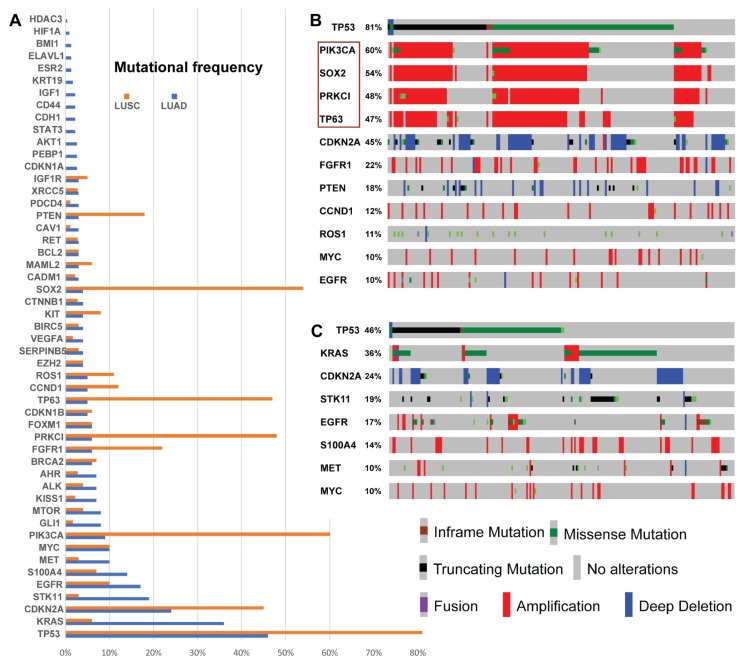
Mutational differences in the 52 driver genes shared by LUAD and LUSC. (**A**) Mutational frequency comparison of the 52 driver genes shared by LUAD and LUSC. (**B**) The sample-based oncoprint mutational profile for the top mutated genes in the TCGA LUAD dataset. (**C**) The sample-based oncoprint mutational profile for the top mutated genes in the TCGA LUSC dataset. LUAD, lung adenocarcinoma; LUSC, lung squamous cell carcinoma; TCGA, The Cancer Genome Atlas.

**Figure 5 biology-12-00357-f005:**
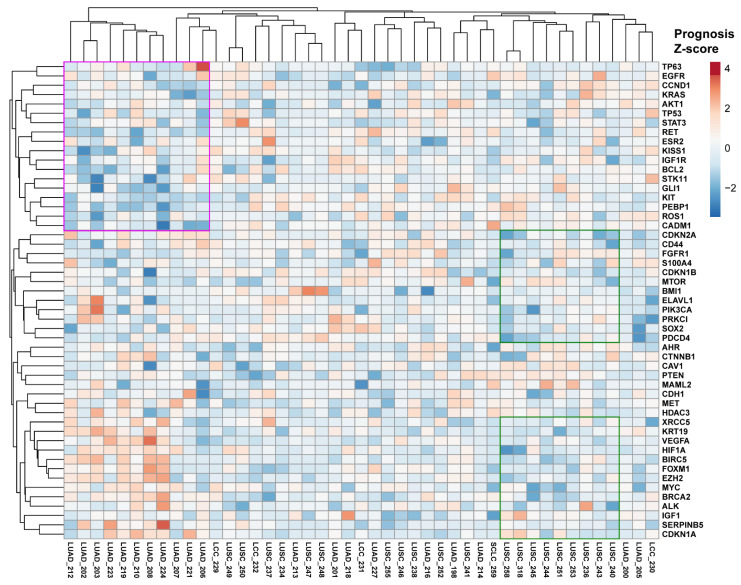
Heatmap of the prognostic Z-scores of 51 genes from 44 LC datasets. Prognostic Z-scores that were obtained from the PRECOG database are represented by the scale bar. The polarity of the prognostic Z-score reflects the direction of association. LCC, large cell carcinoma; LUAD, lung adenocarcinoma; LUSC, lung squamous cell carcinoma.

## Data Availability

All data are available for academic purposes at http://soft.bioinfo-minzhao.org/lcgene (accessed on 12 February 2023).

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
