# Peer review of "A Database of Lung Cancer-Related Genes for the Identification of Subtype-Specific Prognostic Biomarkers"

_biology, 2023, doi:10.3390/biology12030357_

Round 1

Reviewer 1 Report

The overall structure of the paper is good and there is a good effort to put together different resources of omic data related to this pathology. Nevertheless, the paper's novelty is low, as there are already some databases for this cancer pathology as https://doi.org/10.1038/s41388-018-0588-2.  The database has a different kind of information but it seems to lack the possibility to do additional analysis, starting from the data collected into the database. I might expect to have some additional functions, for instance, the possibility to easily do a different kind of analysis as ceRNA or miRNA target interactions, or at least to insert additional links to this kind of database, or to insert new data as single-cell data.

Author Response

Reviewer 1

The overall structure of the paper is good and there is a good effort to put together different resources of omic data related to this pathology. Nevertheless, the paper's novelty is low, as there are already some databases for this cancer pathology as https://doi.org/10.1038/s41388-018-0588-2.  The database has a different kind of information but it seems to lack the possibility to do additional analysis, starting from the data collected into the database. I might expect to have some additional functions, for instance, the possibility to easily do a different kind of analysis as ceRNA or miRNA target interactions, or at least to insert additional links to this kind of database, or to insert new data as single-cell data.

We thank the suggestion. In the updated database, we incorporate ceRNA interaction in the regulatory page as suggested (https://soft.bioinfo-minzhao.org/lcgene/gene_cerna.cgi?gene=999), which will make our database more comprehensive to understand the non-coding RNA regulations. We also update the manuscript accordingly.

“To provide an overview of the potential gene regulation, we also incorporated the experimental validated gene regulation data, including the experimentally validated and predicted competing endogenous RNA (ceRNA) from LncACTdb 3.0.”

Reviewer 2 Report

The authors curated approximately 2,500 LUNG cancer genes from the literature, used downstream analytic methods on these genes, and built a web interface to make these annotations publicly available. This effort necessitates a substantial amount of manual curation of literature, functional annotations and enrichment analysis, and database creation for LUNG cancer research. Researchers working in this field will find the database helpful.

Comments:

1. “In the United States alone, there were about 158,080 LC-linked deaths in 2016.” Why are the authors collecting this statistic in 2016, and not in 2021 or 2022?

2. Figure 1 is not a workflow because there are no arrows indicating the order of the actions.

3. Resolution of Figure 2 is too low

4. Figure 3A, In the Venn diagram, there are genes labelled NSCLC and unknown, how?

5. Section 3.3, the functional impact of some cancer drivers, such as TP53 and KRAS, is frequently achieved by gene mutation or genetic perturbation that does not affect their own expression level. Survival analysis of these genes is required to demonstrate their potential as prognostic biomarkers in sorting patients into groups with different prognostic outcomes.

6. There are numerous typos and grammatical errors throughout the text. It is necessary to perform extensive editing on the English language and style.

Author Response

Reviewer 2

The authors curated approximately 2,500 LUNG cancer genes from the literature, used downstream analytic methods on these genes, and built a web interface to make these annotations publicly available. This effort necessitates a substantial amount of manual curation of literature, functional annotations and enrichment analysis, and database creation for LUNG cancer research. Researchers working in this field will find the database helpful.

Comments:

  1. “In the United States alone, there were about 158,080 LC-linked deaths in 2016.” Why are the authors collecting this statistic in 2016, and not in 2021 or 2022?

We are sorry about it. We updated the statistics to the most recent one and changed the citation accordingly.

  1. Figure 1 is not a workflow because there are no arrows indicating the order of the actions.

We thank the reviewer point this out. We revised our Figure 1 and add the arrows to indicate the workflow.

  1. Resolution of Figure 2 is too low

We are sorry about it. This may be because of the PDF conversion from Tiff. We will work with the editorial team to make sure the figure in high quality.

  1. Figure 3A, In the Venn diagram, there are genes labelled NSCLC and unknown, how?

In our data collection, each entry is a single literature piece of evidence with PubMed ID. Therefore, the same gene may be assigned as NSCLC in some manuscripts but is an unknown cancer subtype in other literature. To avoid any misunderstanding, we revised the manuscript to make this clear.

“It is important to note that different literature may assign different subtypes to the same gene. Consequently, these highly overlapping genes may play consistent roles in numerous subtypes of LC.”

  1. Section 3.3, the functional impact of some cancer drivers, such as TP53 and KRAS, is frequently achieved by gene mutation or genetic perturbation that does not affect their own expression level. Survival analysis of these genes is required to demonstrate their potential as prognostic biomarkers in sorting patients into groups with different prognostic outcomes.

We thank the suggestion. In our figure 4, we run the PRECOG analysis, which is similar to the survival analysis. In the revised manuscript, we follow the suggestion and run the formal survival analysis in terms of mutations associated with 52 genes. By running a survival analysis on 9781 samples from 7590 lung cancer patients in 28 independent studies, we found the 52 genes (including TP53 and KRAS) in Figure 3 are significantly associated with survival outcome (Logrank Test P-Value: 1.130e-4). We added the results as a supplementary figure “Figure S1” in the updated manuscript.  

“In addition, we examined the survival characteristics of 9781 samples from 7590 lung cancer patients in 28 independent studies in order to provide an overview of these genes in the cancer cohort. Figure S1 depicts 52 genes, including TP53 and KRAS, that are sig-nificantly associated with survival outcome (Logrank Test P-Value: 1.130e-4, Figure S1). Therefore, our PRECOG and survival data provide independent evidence that these genes play crucial roles in lung cancer patient survival.”

  1. There are numerous typos and grammatical errors throughout the text. It is necessary to perform extensive editing on the English language and style.

We have asked a native speaker to polish our manuscript. The updated manuscript was in track change format for any further review.

Round 2

Reviewer 1 Report

Although other databases collect different kinds of information on cancer disease, the authors satisfied my requests

Reviewer 2 Report

Accept in present form